



# Drainage of organic soils and GHG emissions: Validation with country data

Giulia Conchedda[1], Francesco N. Tubiello[1]

[1]Statistics Division, FAO, Rome, 00153, Italy

*Correspondence to*: Francesco N. Tubiello (francesco.tubiello@fao.org)

**Abstract.** Drainage of large areas with organic soils was conducted over the past century to free land for agriculture. A significant acceleration of such trends was observed in recent decades in South-East Asia, largely driven by drainage of tropical peatlands, an important category of organic soils, for cultivation of oil palm. This work presents methods and main results of

a new methodology developed for FAOSTAT, whereby the overlay of dynamic maps of land cover and the use of information on histosols allows the production of a global annual dataset of drained area and emissions over a time series, covering the period 1990–2019. This is an improvement over the existing FAO approach, which had produced only a static map of drained organic soils for the year 2000. Results indicate that drained area and emissions increased by 13 percent globally since 1990, reaching in 2019 24 million ha of drained organic soils, with world total emissions of 830 million tonnes of carbon dioxide

($CO_2$) equivalent. Of these totals, the largest contribution was from the drainage of tropical peatlands in South-East Asia, generating nearly half of global emissions. Results were validated against national data reported by countries to the UN Climate Convention and to well-established literature. Overall, the validation yielded a good agreement with these sources. FAOSTAT estimates explained about 60 percent of the variability in official country reported data. The predicted emissions were virtually identical – with over 90 percent of explained variability – to official data from Indonesia, currently the top emitting country

by drained organic soils. Also, calculated emissions factors for oil palm plantations in Indonesia and Malaysia were in the same range and very close to emissions factors derived from detailed field measurements. This validation suggests that the FAO estimates may be a useful and sound reference in support of countries reporting needs. Data are made available as open access via the Zenodo portal (Tubiello and Conchedda, 2020) with DOI 10.5281/zenodo.3942370.

## 1 Introduction

Organic soils are, generally speaking, wet soils ecosystems, characterized by high levels of organic matter, which accumulates in large quantities under the anoxic conditions that exist in the presence of water. They include tropical peatlands, high-latitude bogs and mires. Indeed, while organic soils cover globally a mere 3 percent of the terrestrial land area, they represent up to 30 percent of the total soil carbon, playing an important role in maintaining the earth's carbon balance (FAO, 2020a). Drainage of organic soils releases large quantities of carbon dioxide ($CO_2$) and nitrous dioxide ($N_2O$) into the atmosphere, as a result of

the increased oxidation and decomposition rates of the underlying organic matter once water is removed. These emissions



typically last for several decades after the drainage event, due to the large quantities of organic substrate available. Agriculture is a major cause of drainage of organic soils around the world, and especially since 1990 due to the cultivation of permanent crops such as oil palm. Restoration of degraded organic soils is currently a priority in several countries as part of their greenhouse gas mitigation and ecosystem restoration commitments under the UN climate convention (Leifeld and Menichetti,

2018; Tiemeyer et al., 2020). Measuring current trends, globally and with country detail, is therefore important to identify and quantify existing and fast-developing new hotspots of degradation and to help reduce emissions from drained organic soils in future decades. Estimates of drainage area and greenhouse gas (GHG) emissions from organic soils for the year 2000 were developed by FAO and used by the Intergovernmental Panel on Climate Change (IPCC) for global analysis (Tubiello et al., 2016; Smith et al., 2014). That preliminary work was based on the geospatial overlay of two static maps, one for land cover,

indicating presence of agriculture, and one for soil characteristics, indicating presence of organic soils, through the use of histosols as proxy. This paper describes additional methodological developments made possible by the availability of time dependent land cover maps, resulting in the production, for the first time, of estimates over a complete time series (1990–2019).

## 2 Material and Methods

Organic soils are characterized by high concentrations of organic matter. They mostly develop under poorly drained, wetland conditions and are found at all altitudes, with the vast majority occurring in lowlands (Rieley and Page, 2016). Peatlands are an important type of organic soils (Page et al., 2011; IPCC, 2014a). According to IPCC (2006), organic soils can be largely identified with the *histosols* group of the FAO-UNESCO classification. FAO and Wetlands International (2012) indeed described *histosols* as soils that develop in (predominantly) "moss peat in boreal, arctic and subarctic regions, *via* moss peat,

reeds/sedge peat and forest peat in temperate regions to mangrove peat and swamp forest peat in the humid tropics". Common names for *histosols* are `peat soils', `muck soils', `bog soils' and `organic soils' (FAO et al., 1998). In this work, we follow IPCC guidelines and identify organic soils with histosols. Cropland and grassland organic soils are drained permanently or semi-permanently, as well as regularly limed and fertilized, to permit annual or permanent crop cultivation, including tree plantations, or to support livestock grazing. For these reasons, the presence of cultivated crops or grazing animals on organic

soils (Noble et al., 2018) —which have otherwise low productivity—may be associated with drainage, causing N and C losses and overall degradation (Martin et al., 2013). Peat emissions are unique in that they continue emitting for long periods after the initial drainage (FAO, 2020).

Area of drained organic soils and associated greenhouse gas (GHG) emissions were estimated following default Tier 1

methods of the IPCC (2006) over IPCC land use classes *Cropland* and *Grassland* (corresponding to FAO land use classes ''Cropland'' and ''Land under permanent meadows and pastures''). This methodology was already applied within



FAOSTAT (Tubiello et al., 2016) and was extended herein by introducing a time-dependent component, as follows in Eq. (1):

$$Emissions_{y,i,j} = \sum_{y,j} A * EF_{i,j,k}$$ Eq. (1)

where:

$Emissions_{y,i,j}$ = Emissions for year $y$ of greenhouse gas $i$ = N$_2$O, CO$_2$ over land use type $j$ = cropland, grassland;

$\Sigma_{y,j} A$ = Total area for year $y$ of drained organic soils under land use type $j$ = cropland, grassland;

$EF_{ijk}$ = Emissions Factors, emissions per unit area of drained organic soils of greenhouse gas $i$, land use type $j$ and climatic zone $k$;

y = Years in the period 1992–2018 as yearly time-steps representing time-dependent land cover maps;

$k$ = Boreal, temperate, tropical climatic zones, following IPCC (2006).

At pixel level, the work we carried out included the use of a geospatially-detailed map of organic soils, annual maps of land cover, a livestock density map and a map of climatic zones. The area drained for cropland and grassland organic soils represent the time-dependent components of Eq. (1). They were calculated as follows:

$$A_{cropland,y} = LU_{cropland,y} * WMS_{histosols}$$ Eq. (2)

$$A_{grassland,y} = LU_{grassland,y} * WMS_{histosols} * LDR_{>0.1}$$ Eq. (3)

Where:

A$_{cropland\_y}$ ; A$_{grassland\_y}$ = Area of drained organic soils on cropland and grassland, for the year $y$ obtained as the overlay of

LU$_{cropland,y}$ ; LU$_{grassland,y}$ = Area, for the year $y$ under IPCC and FAO land use class ''cropland'' and "grassland", derived from land cover classes (cropland or grassland) in global land cover maps of the year $y$;

WSM$_{histosols}$ = Area with soil type $histosols$ from the Harmonized Soil Map of the World (FAO and IIASA, 2012). Following IPCC (2006), $histosols$ are used as proxy for organic soils;

LDR$_{>0.1}$ = For grassland organic soils only, area with livestock density of ruminants (in livestock units) above a defined threshold, derived from global maps of the FAO Gridded Livestock of the World (Robinson et al., 2014), to identify grazed grassland.

The IPCC basic methodology for carbon (C) and nitrous oxide (N$_2$O) emissions estimates from organic soils assigns an annual EF (i.e. quantity of gas emitted per ha and per year per unit of activity data) associated with the loss of these gases following

the drainage for agriculture. Drainage stimulates the oxidation of organic matter previously built up under a largely anoxic

environment. The rates of emissions are influenced by climate, with warmer climates accelerating the processes of oxidation of soil organic matter hence causing higher emissions than in temperate and cooler climates. The emission factors by gas thus are climate-dependent. In this methodology, we spatialized the relevant IPCC emission factors following a global map of climatic zones (JRC, 2010) to produce global maps EF for the two gases. The pixel-computations then multiply the area of drained and managed organic soils from Eq. (2) and Eq. (3) above by global maps of emission factors to derive estimates of

annual $N_2O$ and $CO_2$ emissions by pixel as summarized in Eq. 1.

As described in Tubiello et al., (2016), the approach is based on reclassification tables to extract the proportions of cultivated and grassland area from the yearly land cover maps. When all input layers overlap, the underlying assumption is that of an equal likelihood within each pixel to find cultivated (or grassland) area and organic soils. Operationally, the methodology multiplies the area of organic soils in the pixel by the area of the pixel that is cropped or has grassland cover. In this way, we

derived by pixel the area of organic soils that is drained for agricultural activities. Organic soils must be indeed be drained to allow for crop cultivation activities. In the case of grassland, livestock grazing beyond the carrying capacities of organic soils leads instead to peat degradation and drainage. The following sections provide more details about the information necessary to implement the computations above.

## 2.1 Soils

Information on the geographical distribution of *histosols*, for use in the term WSM of Eq. (2) and Eq. (3) above, was derived from the Harmonized World Soil Database (HWSD v 1.2), a raster dataset with a nominal resolution of 30 arc second on the ground (corresponding approximately to 1 x 1km at the equator) published in 2012 by FAO and the International Institute for Applied Systems Analysis (IIASA). The HWSD compiles more than 40 years of soil information from several sources worldwide, re-classified and harmonized according to the FAO-UNESCO classification. The standardized structure of the

HSWD v 1.2 allows displaying and querying the composition in terms of soil units and of soil parameters such as the organic carbon content, the pH, or the water storage capacity. The HSWD dataset was queried to extract values representing the percentage of the pixel area that contains histosols, as either dominant or secondary soil type (Fig. 1).

## 2.2 Land cover and land use

Information on the area extent of IPCC categories *cropland* and *grassland* for use in terms $LU_{cropland}$ and $LU_{grassland}$ in Eq. (2)-(3) was taken from the land cover maps produced by the Catholic University of Louvain (UCL) Geomatics (UCL Geomatics, 2017), produced under the Climate Change Initiative of the European Spatial Agency (ESA CCI, 2020) and hereinafter referred to as CCI LC maps. The CCI LC maps were first released in April 2017 as 24 global annual and consistent land cover maps covering the period 1992 to 2015 (UCLouvain Geomatics, 2017). At the end of 2019 and in the framework of the European

Copernicus Climate Change Service (C3S, 2019), three new global land cover (LC) products were released for the years 2016, 2017 and 2018 that are consistent with earlier maps (Fig. 2).

The long-term consistency of this dataset, yearly updates and high thematic detail on a global scale make it uniquely suitable to observe and assess changes in area drained and GHG emissions from organic soils. The CCI LC maps contain information

for 22 global land cover classes, based on the FAO Land Cover Classification Systems (Di Gregorio, 2005), with a spatial resolution of approximately 300m.

The land cover maps (1992–2018) were used to assign to each pixel the proportion of its area under relevant land cover categories. This information was combined to provide proxy information on the proportion of pixel area under land cover / land use classes *cropland* and *grassland* (Tables 1 and 2).


### 2.3 Livestock

Information on the spatial distribution of livestock for use in estimating the term LDR in Eq. (3) above, was taken from the Gridded Livestock of the World (GLW)(Robinson et al., 2014), providing geospatial data on the density of three ruminants species: cattle, sheep and goats (Fig. 3). Animal numbers by pixel were first converted in livestock units (LSU)(FAO, 2011),

and pixels with values higher than 0.1 (Critchley et al., 2008; Worrall and Clay, 2012) selected for use in Eq. (3).

### 2.4 Climatic Zones and emission factors

As discussed above, pixel-level climatic information for use in terms $EF_{ijk}$ in Eq. (1), was derived from a map of climatic zones. The Joint Research Centre (JRC) of the European Commission, developed this spatial layer in line with IPCC specifications based on latitude and elevation of each pixel (Figure 4).

Default IPCC emissions factor by land use and gas (Table 3) were then assigned by pixel to each climatic zone and three additional geospatial layers were produced to cover possible combinations of EFs (
Figure 5). As one country may encompass more than one climatic zones, when emissions are aggregated at national level, the resulting emissions factors represent weighted averages of the various EFs assigned at pixel level. In computations, $CO_2$-C losses are converted to $CO_2$ values multiplying by 44/12 while $N_2O$ emissions are calculated multiplying $N_2O$-N values by

150 44/28.

### 2.5 Data Availability: Structure of the FAOSTAT datasets on ''Organic Soils" and online access

Results from the spatial computation are aggregated at national level for 101 countries and 4 territories, representing the subset of FAOSTAT countries and territories where organic soils are present. Statistics are disseminated in three separate FAOSTAT domains (FAO 2020b,c,d), over the period 1990–2019, in line with country reporting requirements to the Climate Convention



and following the IPCC (2006). Namely, statistics are disseminated by gas and land use class: emissions of $N_2O$ on cropland and grassland are disseminated under the domain Cultivation of Organic soils (FAO, 2020b) of FAOSTAT Emissions-agriculture; whereas emissions of $CO_2$ on Cropland (FAO, 2020c) and Grassland (FAO, 2020d) are disseminated within the FAOSTAT Emissions-Land use domain. As part of ongoing efforts to provide users with reliable and transparent data, the complete spatial dataset that underlies FAOSTAT statistics will be also disseminated through FAO new maps catalog (FAO,

2020e). Under the dataset, Cultivation of Organic soils, $N_2O$ emissions are also disseminated in $CO_2eq$ by applying three different sets of Global Warming Potential (GWP) coefficients (100-year time horizon) from the IPCC assessment reports: *a)* IPCC Second Assessment Report (IPCC, 1996); *b)* IPCC Fourth Assessement Report (IPCC, 2007); and *c)* IPCC Fifth Assessment Report (IPCC, 2014b). All data are also available at Zenodo as open access (Tubiello and Conchedda, 2020) with DOI 10.5281/zenodo.3942370. They and can be downloaded at https://zenodo.org/record/3942370#.XxWJjygzbIU.

**2.6 Limitations and uncertainty**

Previous work had estimated the uncertainty of our estimates at ±40% for the area information and an uncertainty range (−14%, +166%) for the emission estimates. These uncertainties, valid at pixel level, were assumed to also characterize the nationally-aggregated values (Tubiello et al., 2016). Furthermore, the new methodology developed herein may result in some cases in reduction in the drained area during the 30 years of the analysis (see Appendix A, Table A1). In such cases, the pixel-level

proportions that are applied to identify the cropland and grassland cover, have detected corresponding changes in land cover. The scale of the analysis prevents however to understand whether these changes actually happened in the area of organic soils thus resulting in a rewetting of the drained peats or are instead an artefact of the spatial methods. The IPCC Wetlands Supplement introduced already in 2014 additional methodological guidance, with a specific focus on the rewetting and restoration of peatland that was not included in the 2006 Guidelines (IPCC, 2014a). Limited country-specific activity data on

rewetting prevented however implementing the Supplement refined methods in the FAOSTAT dataset.

**3 Main results: Global trends**

In 2019, nearly 25 Mha or about 7.5 percent of the 328 Mha of worldwide *histosols* had been drained for agriculture with a limited increase since 1990. Data suggest that the largest extent of organic soils in Northern America and Eastern Europe have undergone little changes during the past decades likely because these peats have been drained for agriculture already for many

centuries (Joosten and Clarke, 2002). The drainage of organic soils is instead a more recent phenomenon in South East Asia. In this region, the drained area grew by 5 percent points since 1990 and in 2019 more than 26 percent of the original organic soils were already drained. Asia is on average the region with the highest share of drained *histosols* (30 percent) while, at sub-regional level Western Europe had over two thirds of its organic soils that were drained already in 1990. In 2019, among countries where the area of *histosols* is above 1Mha (see Annex A, Table A1), the larger proportions of drained organic soils





were in Mongolia (over 80 percent), Germany (75 percent), Poland (60 percent), United Kingdom and Belarus (about 50 percent). Nearly one fourth of the original extent was drained in Indonesia and Zambia and 30 percent in Malaysia.

In 2019, Indonesia had the largest area of drained organic soils (newly 5Mha), followed by the Russian Federation (about 1.9 Mha) and the United States of America (nearly 1.6 Mha). Among these top ten countries, Indonesia and Malaysia also registered the largest relative increases in area drained since 1990 (+5 and +10 percent for Indonesia and Malaysia,

respectively).

Global GHG emissions from drained organic soils were 833 Mt $CO_2$eq. In 2019, emissions were 13 percent higher when compared to 1990 and 10 percent higher when compared to 2000 (Figure 6). This value represented almost 8 percent of total agriculture and related land use emissions.

In 2019, $CO_2$ and $N_2O$ gas contributed 87 percent and 13 percent of global emissions. Grassland organic soils were

responsible for about 10 percent of all emissions while the vast majority was due to the drainage for cropping. These relative contributions have changed little since 1990 (Appendix A, Table A2).

In 2019, three-fourths of the global emissions from organic soils were from only 11 countries (Figure 7), Malaysia and Indonesia together were responsible for nearly half (47 percent) of total emissions.

## 4. Results: Data Validation

The FAOSTAT estimates of the extent of organic soils, which are used as input to Eq. (2)–(3), were compared to published data at country, regional and global level. Resulting emissions and emissions factors for oil palm plantations are also included to validate FAOSTAT results.

### 4.1 Area of organic soils and peatlands

Comparison of the extent of drained organic soils is hindered by a number of factors, including the fact that the FAOSTAT

data refers to area of organic soils, while a majority of published studies has focused on area of peatlands. The FAOSTAT global estimates of 3.3 million square kilometers ($Mkm^2$) of organic soils (*histosols*) were 25 percent smaller than the published range of 4.0-4.3 $Mkm^2$ of peat soils. This is consistent with statements by Xu et al. (2018), who highlighted that *histosols* tend to underestimate areas in tropical swamp-forested peatlands. At regional level, FAOSTAT data agreed well with the most recent estimates of Xu et al. (2018) and mean estimate from Immirzi et al. (Table 5). In addition, while acknowledging the

large differences existing between published estimates by regions, FAOSTAT estimates remained consistently within the observed ranges More specifically, FAOSTAT estimates of area of organic soils for North America was 1.3 $Mkm^2$ vs. a published range of 1.3 – 1.9 $Mkm^2$; for Asia, FAOSTAT estimated 0.3 $Mkm^2$ vs. a range of 0.3 – 1.5 $Mkm^2$; for Europe, FAOSTAT estimated 1.5 $Mkm^2$ vs. a range of 0.6 – 1.9 $Mkm^2$; for Africa, FAOSTAT estimated 0.07 $Mkm^2$ vs. a range of 0.05–0.2 $Mkm^2$; for South America, FAOSTAT estimated 0.1 $Mkm^2$ vs. a range of 0.09 – 0.5 $Mkm^2$; and for Oceania,

FAOSTAT estimated 0.05 $Mkm^2$ vs. a range of 0.00 – 0.07 $Mkm^2$.





We continued the validation analysis by comparing FAOSTAT estimates to published data for about 60 tropical countries, compiled from the widely recognized meta-analysis of Page et al. (2011), and for the same set of countries to values computed using a recent map of tropical peat distribution (Gumbricht et al., 2017)(Appendix B, Table B1). In 2017, Gumbricht and associates published new estimates of wetland and peatland areas, depths and volumes. The expert system approach is based

on three biophysical indices related to wetland and peat formation: (1) long-term water supply exceeding atmospheric water demand; (2) annually or seasonally water-logged soils; and (3) a geomorphological position where water is supplied and retained. At the aggregate level—the sum of area of organic soils in countries covered by Page et al. (2011)—the extent of tropical organic soils estimated by FAOSTAT was 0.43 Mkm2 , which compared well with the value of 0.44 Mkm$^2$ of Page, both about a third of the total in Gumbricht et al. (2017) (1.37 Mkm$^2$).

At country level, FAOSTAT estimates agreed well with data published by Page et al. (2011) (R=.677, $p$<0.001). For one percent increase in the area of histosols, the log-transformed model shows about 5.5 percent increase in the area of peat as mapped by Page and colleagues (R$^2$=.458) (Fig. 8). The largest differences were found in countries in South and Central America—where special formations at high altitudes and dry conditions, known as *paramos*, may be poorly captured as *histosols* (Lähteenoja et al., 2012). FAOSTAT and Page et al. (2011) data were in very close agreement for key global

contributors in Southeast Asia, Indonesia (0.20 vs. 0.21 Mkm$^2$) and Malaysia (0.02 vs. 0.03 Mkm$^2$).

FAOSTAT country-level estimates were, albeit to a lesser degree, also in good agreement with those obtained by aggregating geospatial information from Gumbricht et al. (2017) (R=0.541, $p$<0.0005). FAOSTAT histosols however explains only partially the variability in the peat area as mapped by these authors. (R$^2$=.293) (Fig. 9). For one percent increase in the area of histosols, the log-transformed model shows a 8 percent increase in the area of peat as mapped by Gumbricht et al. Significant

differences between FAOSTAT and this second, independent set of observed data included Brazil, where Gunbricht and colleagues (2017) estimated 0.31 Mkm$^2$ of organic soils, nearly forty times the area estimated in FAOSTAT and more than ten times the area published in Page et al. (2011); Peru, where Gumbricht et al. (2017) indicate some 0.08 Mkm$^2$, twice the FAOSTAT estimates and 0.02 Mkm$^2$ more than Page et al. (2011); and the Democratic Republic of Congo, where the new peatland map suggests a significant presence of organic soils (0.12 Mkm$^2$), consistently with recent studies (Dargie et al.,

2017), while FAOSTAT estimated only 240 km$^2$ and Page et al. (2011) less than 3000 km$^2$.

The use of observed or estimated data is hampered by the wide uncertainties that still exist in defining, mapping and measuring actual extent of peatland throughout the world. To date, no globally accepted definition of peatlands exists. To this end, ongoing international efforts such as the Global Peatlands Initiative (2020) are expected to improve and consolidate current knowledge.

**4.2 Validation with country data reported to the Climate Convention of the United Nations**

The FAOSTAT data uses Eq. (2)–(3) above to overlay information on organic soils extent with information on land use and other geospatial characteristics, to estimate the drainage area of organic soils due to agriculture (Tubiello et al., 2016). These were in turn used as input to estimate resulting GHG emissions. We used data reported by countries to the UN Framework



Convention on Climate Change (UNFCCC) for validation of these FAOSTAT estimates. We looked both at data from the National Greenhouse Gas Inventories of the Annex I Parties and to the national communication from Indonesia, a top emitter

country.

### 4.2.1 Annex I parties

UNFCCC data were available for thirty-eight countries belonging to the Annex I parties to the climate Convention. These represent developed countries, mostly located in temperate and boreal zones of the world. First, we compared data on the area drained (activity data), which allowed to test assumptions underlying the use of Eq. (2)–(3) above. FAOSTAT country-level

estimates were in good agreement with those officially reported by countries to the UNFCCC ($R^2$=0.57) of area drained of organic soils (Fig. 10). At regional level, FAOSTAT predicted a total of about 14 Mha of drained organic soils for Annex I parties, versus country reported figures of nearly 12 Mha for the last inventory in 2017 (Appendix B, Table B2). On the one hand, estimates in several countries with significant contributions were well in line with national reporting, including the United States of America (1.5 *vs* 1.4 Mha); Belarus (close to 1.4 Mha in both cases); Germany (1.1 *vs* 1.2 Mha). On the other,

significant differences were found in Poland (1.0 *vs* 0.7 Mha) and in the United Kingdom (1.3 *vs* 0.3 Mha). Wide differences also characterized two countries with major organic area extent, specifically the Russian Federation (1.8 *vs* 4.3 Mha) and Canada (1.3 *vs* 0.2 Mha). In these latter cases, differences have however opposite directions. FAOSTAT estimates were much larger than country reported data in Canada but smaller in the Russian Federation.

For the same set of UNFCCC countries as above, we also compared $N_2O$ emissions, which are reported by countries under the

IPCC sector Agriculture. C fluxes from the drainage of organic soils are reported by Annex I countries under Land Use, Land Use Change, and Forestry (LULUCF) categories. In the inventories, relevant reporting categories are 4.B.1 "Cropland Remaining Cropland"; 4.B.2 "Land Converted to Cropland" and 4.C.1 "Grassland Remaining Grassland"; and 4.C.2 "Land Converted to Grassland". Data for carbon are much sparser than for $N_2O$ emissions possibly due also to complexity in reporting (Berthelmes et al., 2015). Beside the differences in activity data (area drained) that were observed earlier, differences may also

due to countries applying Tiers higher than the default methodology applied in FAOSTAT as well as to the definition of types of land use causing drainage.

As for the area drained, FAOSTAT emissions estimates were also in good agreement with data officially reported to the UNFCCC ($R^2$=0.553) (Fig.11), but with FAOSTAT consistently overestimating country data. At regional level, FAOSTAT predicted total emissions of 184 kt $N_2O$ for Annex I parties, versus country reported figures of 143 kt $N_2O$ (Appendix B, Table

B2). Estimates of annual emissions in several countries with significant contributions were well in line with national reporting, including the United States of America (20 *vs* 27 kt $N_2O$); Belarus (19 *vs* 18 kt $N_2O$); Germany (14 *vs* 10 kt $N_2O$); and Ukraine (8 *vs* 6 kt $N_2O$). At the same time, significant differences characterized two countries with major organic area extent, specifically the Russian Federation (23 vs. 54 kt $N_2O$) and (16 vs. 0.2 kt $N_2O$).

FAOSTAT results are in line with other independent assessments, for instance a study for countries in the Baltic region

(Barthelmes et al., 2015) suggested that the area and emissions from drained organic soils are often underestimated in



UNFCCC reporting. In a recent paper, Tiemeyer et al. (2020) applied for Germany a refined methodology for organic soils in national GHG inventories using detailed activity data and IPCC Wetlands EFs. For a similar extent of drained organic soils (about 12 Mha) as in FAOSTAT, their emissions estimates from cropland and grassland drained organic soils were 45 Mt $CO_2$eq, about three-fold FAOSTAT results (14 Mt Mt $CO_2$eq). This suggests, that even FAOSTAT estimates may not fully

grasp the potential for mitigation from the rewetting of drained organic soils.

### 4.2.2 Non-Annex I parties

Over fourty percent of the global emissions from the agricultural drainage of organic soils is generated in Indonesia and Malaysia. In addition, these two countries have contributed the most to emissions increases since 1990 (FAO, 2020f) (Fig. 12).

We compared FAOSTAT estimates of GHG emissions from the drainage of organic soils in Indonesia to those reported by the country to the UNFCCC, over the period 2000–2016 in their second Biennial Update Report (BUR), submitted to the Climate Convention on December 2018. National reported data were based on a national map of peatland and on refined EFs from the Wetlands Supplement (IPCC, 2014a) and reported as distinct category "peat decomposition" under Forestry and Other Land Use (FOLU) emissions. Results indicated good agreement between FAOSTAT estimates and nationally reported

data. The average FAOSTAT GHG emissions over 1990–2016 were 281 Mt $CO_2$eq *vs* the 304 Mt $CO_2$eq reported by Indonesia to the UNFCCC. Both series have a similar upward trend and their agreement extended over the entire time series ($R^2$= 0.9446), albeit with increasing separation in the most recent years. This represents an additional validation of Eq. (2) and Eq. (3) in the methodology addressing the issue of time dependence in drainage data (Fig. 13). As FAOSTAT only includes drainage for agriculture, part of the differences may be due to the types of land use for which the BUR reports drainage and emissions, as

the BUR possibly includes drainage under forestry.

### 4.2.3 Emissions factors for palm oil plantations

The establishment of new oil palm plantations is recognized as main driver for the drainage of tropical peatlands in Indonesia and Malaysia (Hojer et., 2010; Hooijer et al., 2012;  Miettinen et al., 2012; Dohong et al., 2018; Cooper et al., 2020; FAO, 2020f). As a supplementary validation of Eq. (1)–(2), we combined spatially the cropland land use layer to an additional map

of tree plantations produced by Petersen et al. (2016) (Fig. 14). These authors mapped the distribution of different types of plantations  including oil palm –for the years 2012–2014 in seven countries: Brazil, Cambodia, Colombia, Indonesia, Liberia, Malaysia, and Peru, using satellite imagery and extensive field validation, particularly for Indonesia and Malaysia. This additional analysis allows to compare FAOSTAT resulting emissions factors (*i.e.* the emissions per unit area of oil palm on drained organic soils), to those in established literature.


Peterson and associates estimated that oil palm plantations in Indonesia and Malaysia covered 11.7 Mha and 5.3 Mha in 2014, respectively. These were consistent with 2014 FAOSTAT statistics on oil palm harvested area, indicating 8.1 Mha and 4.7





Mha respectively for the two countries. Of these, based on our analysis of the crop drained soils and of the plantations map, about 9 percent (Indonesia) and 4 percent (Malaysia) were located in drained organic soils (Table 6). Together, the tree plantations mapped by Petersen and colleagues were responsible for almost half of the 2014 emissions from cropland organic soils in Indonesia.

In Malaysia, the relative contribution of oil palm plantations to the area drained for cultivation was even as nearly half of total area of crop organic soils and emissions in this country was due to oil palm plantations. Other types of plantations contributed an additional 10 percent to area drained and emissions, which suggests the contribution of annual and temporary crops to peat conversion in this country may less important than in Indonesia.

The EFs for oil palm plantations derived from the analysis was around 78 $CO_2$eq ha$^{-1}$ yr$^{-1}$ in the two countries, and in close agreement with published estimates (Table 10). Available literature is largely based on direct measurements and typically analyses the influence of the depth of drainage, soil subsidence rates, soil moisture and the period since the initial drainage and establishment of the oil palm plantations. Corresponding values range from minimum average losses of 13 t $CO_2$e ha$^{-1}$ yr$^{-1}$ as in Hashim et al., (2018) to a maximum value of 117 t $CO_2$e ha$^{-1}$ yr$^{-1}$ as in Matysek et al. (2018) and recent study by Cooper et al. (2020). FAOSTAT estimated EF is therefore very close to the average value from the selected studies (73 t $CO_2$eq ha$^{-1}$ yr$^{-1}$). This additional validation confirms that our methodology is compatible with most relevant and well-established estimates of a major source of emissions from drained organic soils in South East Asia and suggests that FAOSTAT estimates may be equally applied to other tropical countries.

## 5 Conclusions

Organic soils are a rich carbon pool and their drainage for agriculture has important impacts on the global carbon cycle. FAOSTAT statistics on greenhouse gas emissions relative to the drainage of organic soils were updated for the period 1990–2019 based on geospatial computation and pixel-level application of default Tier I method of the Intergovernmental Panel on Climate Change (IPCC). In line with country reporting requirements to the Climate Convention, and following the IPCC, statistics are disseminated by gas ($N_2O$ and $CO_2$) and land use classes, cropland and grassland in three separate FAOSTAT domains. These FAOSTAT statistics represent the only avaialble global dataset in the world today showing country, regional and global time series on drained organic soils. Efforts are also in progress to disseminate publicly the underlying spatial data.

In 2019, FAOSTAT estimated that nearly 25 million ha of organic soils were drained from agriculture and were responsible for 833 million tonnes of $CO_2$eq. This was about 8 percent of total agriculture and related land use emissions in that year. About half of the greenhouse gas emissions was due to the drainage of organic soils in South Eastern Asia and particularly Indonesia and Malaysia.

We validated methods and results by comparing data reported by countries to the United Nations Climate Convention on Climate Change including in the comparison both data from developed countries of the Annex I group and Indonesia, a top emitter country for drained organic soils. For this latter country we also validated with additional analysis the resulting emission





factor for oil palm plantations, a major driver of the emissions in South East Asia. FAOSTAT statistics are well aligned with country reported data and the most established literature. Overall, FAOSTAT statistics explained about 60 percent of the variability in official reported data. However, in Indonesia, the top emitter country by drained organic soils, as well as in many developed countries FAOSTAT statistics yielded an even higher agreement and proved a robust estimator of country official data. This suggests that the FAOSTAT database may provide a useful global reference in support of countries reporting

requirements while national capacities are being developed.

Following guidelines of the Intergovernmental Panel on Climate Change, FAOSTAT statistics are computed applying *histosols* as proxy for organic soils. However, wide uncertainties still remain as whether organic soils may fully capture the dynamics in peat distribution and related emissions particularly in tropical countries. FAO ongoing efforts under the Global Peatland Initiative are expected to provide advancements for mapping and monitoring peatlands worldwide.


**Acknowledgments.** This work was made possible by regular funding provided to FAO by its member countries. We are grateful to staff of the FAO Statistics Division for overall support, and in particular to Griffiths Obli-Layrea for UNFCCC data provision and Amanda Gordon for FAOSTAT maintenance and dissemination. The views expressed in this publication are those of the authors and do not necessarily reflect the views or policies of FAO.




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



**Tables**


Table 1**. Proportion of area of relevant CCI-LC pixels corresponding to land use *cropland***

| CCI-LC Class Code | CCI-LC Land Cover Class Label | Cropland Assigned pixel area (%) |
|---|---|---|
| **10** | Cropland rainfed | |
| *11ᵃ* | *Cropland, rainfed, herbaceous cover* | 85% |
| *12ᵃ* | Cropland, *rainfed, tree or shrub cover* | |
| **20** | Cropland, irrigated or post-flooding | |
| **30** | Mosaic cropland (>50%) /natural vegetation (tree, shrub, herbaceous cover) (<50%) | 60% |
| **40** | Mosaic natural vegetation (tree, shrub, herbaceous cover) (>50%) / cropland (<50%) | 40% |

ᵃ Corresponding to more detailed classification (Level 2) in CCI-LC maps but with limited geographical availability.






Table 2. **Proportion of area of relevant CCI-LC pixels corresponding to land cover *grassland***

| CCI-LC Class Code | CCI-LC Land Cover Class Label | Grassland Assigned pixel area (%) |
|---|---|---|
| **130** | Grassland | 100% herbaceous cover |
| **140** | Lichens and mosses | |
| **120** | Shrubland | |
| ***121ᵃ*** | *Evergreen shrubland* | |
| ***122ᵃ*** | *Deciduous shrubland* | |
| **30** | Mosaic cropland (>50%) / natural vegetation (tree, shrub, herbaceous cover) (<50%) | 30% (20% herbaceous + 10% shrub cover) |
| **40** | Mosaic natural vegetation (tree, shrub, herbaceous cover) (>50%) / cropland (< 50%) | 40% (20% herbaceous + 20% shrub cover) |
| **100** | Mosaic tree and shrub (>50%) / herbaceous cover (<50%) | 55% (30% herbaceous + 35% shrub cover) |
| **110** | Mosaic herbaceous cover (<50%) / tree and shrub (>50%) | 80% (60% herbaceous + 20% shrub cover) |
| **10** | Cropland rainfed | 5% natural herbaceous cover |
| ***11ᵃ*** | *Herbaceous crops* | |

ᵃ Corresponding to more detailed classification (Level 2) in CCI-LC maps but with limited geographical availability.




Table 3. **IPCC default emission factors assigned, by gas and climatic zone[a]**

| Climatic zone | EF: CO2-C (t C ha⁻¹ yr⁻¹) | | *Uncertainty* | EF: $N_2O$-N (kg ha⁻¹ yr⁻¹) | | *Uncertainty* |
|---|---|---|---|---|---|---|
| | **Cropland** | **Grassland** | | **Cropland** | **Grassland** | |
| **1. Warm Temperate Moist** | 10 | 2.5 | | *8* | | |
| **2. Warm Temperate Dry** | | | | | | |
| **3. Cool Temperate Moist** | | | | | | *Range 2 – 24* |
| **4. Cool Temperate Dry** | | | | | | |
| **5. Polar Moist** | | | | | | |
| **6. Polar Dry** | 5 | 0.25 | | 8[b] | | |
| **7. Boreal Moist** | | | ± *90%* | | | |
| **8. Boreal Dry** | | | | | | |
| **9. Tropical Montane** | | | | | | |
| **10. Tropical Wet** | 20 | 5.0 | | 16 | | *Range 5 – 48* |
| **11. Tropical Moist** | | | | | | |
| **12. Tropical Dry** | | | | | | |

[a] Adapted from table 5.6; Table 6.3 (for $CO_2$) and Table 11.1 (for $N_2O$) of IPCC (2006). [b] Default value not included in IPCC (2006), assumed equal to EF values for Cool Temperate zones.





Table 4. **Global Warming Potentials (GWP) relative to CO₂ (dimensionless)**

| Greenhouse gas | GWP SAR (IPCC 1966) | GWP AR4 (IPCC, 2007) | GWP AR5 (IPCC, 2014) |
|:---:|:---:|:---:|:---:|
| $N_2O$ | 310 | 298 | 265 |





Table 5. **Comparisons of published global and regional estimates for area of peat / organic soils (km²)[a]**

|  | Immirzi et al. (1992) mean | Lappalainen (1996) best estimate | Joosten and Clarke (2002) maximum | Xu et al., (2018) | FAOSTAT 2020 |
|---|---|---|---|---|---|
| **North America** | 1,710,470 | 1,735,000 | 1,860,000 | 1,339,321 | 1,311,595 |
| **Asia** | 338,208 | 1,119,000 | 1,523,287 | 283,861 | 258,686 |
| **Europe** | 1,784,887[b] | 957,000 | 617,492 | 1,867,658[c] | 1,501,696[c] |
| **Africa** | 49,765 | 58,000 | 58,534 | 187,061 | 72,445 |
| **South America** | 86,271 | 102,000 | 190,746 | 485,832 | 99,860 |
| **Oceania** | 230 | 14,000 | 8,009 | 68,636 | 45,095 |
| **Total** | **3,969,831** | **3,985,000** | **4,258,068** | **4,232,369** | **3,289,377** |

[a] Adapted and extended from Rieley and Page (2016). [b] Immirzi et al (1992) estimates for Europe include the Soviet Union. [c]Xu
et al., 2018 and FAO estimates for Europe include the Russian Federation.



Table 6. **Contribution of oil palm and other tree plantations to area drained and emissions from crop organic soils, Indonesia and Malaysia in 2014**

| | Mt $CO_2$ | Mt $N_2O$ | Mt $CO_2eq$ | Share (%) of emissions from crop organic soils, by plantation type |
|---|---|---|---|---|
| **Indonesia** | | | | |
| Oil palm | 66.8 | $23*10^{-3}$ | 72.8 | 23% |
| Oil palm mixed | 10.1 | $3*10^{-3}$ | 19.5 | 6% |
| Other tree plantations | 58.2 | $20*10^{-3}$ | 63.5 | 20% |
| **All tree plantations** | **76.9** | **$26*10^{-3}$** | **155.9** | 49% |
| **Malaysia** | | | | |
| Oil palm | 16.3 | $6*10^{-3}$ | 17.8 | 38% |
| Oil palm mixed | 3.8 | $1*10^{-3}$ | 4.1 | 9% |
| Other tree plantations | 4.2 | $1*10^{-3}$ | 4.6 | 10% |
| All tree plantations | 24.3 | $8*10^{-3}$ | 26.4 | 57% |




Table 7**. Comparison of EFs for oil palm plantations on organic soils from established literature and combined FAOSTAT / Petersen et al. (2016) spatial data**

| Source | $CO_2eq\ ha^{-1}\ yr^{-1}$ |
|---|---|
| **Published studies** | |
| Page et al., 2011 | 86−100 |
| Hojer et al., 2012 | 78 |
| Agus et al., 2013 | 43 |
| Couwenberg and Hooijer, 2013 | 66 |
| Hashim et al. 2018 | 13−53 |
| Matysek et al., 2018 | 86−117 |
| Cooper et al., 2020 | 70−117 |
| FAOSTAT / Petersen et al., 2016 | 78 |




**Figures**








**Figure 1. Global distribution of histosols, percentage of pixel area**







**Figure 2. Global land cover, 1992–2018 composite information from CCI-LC maps**




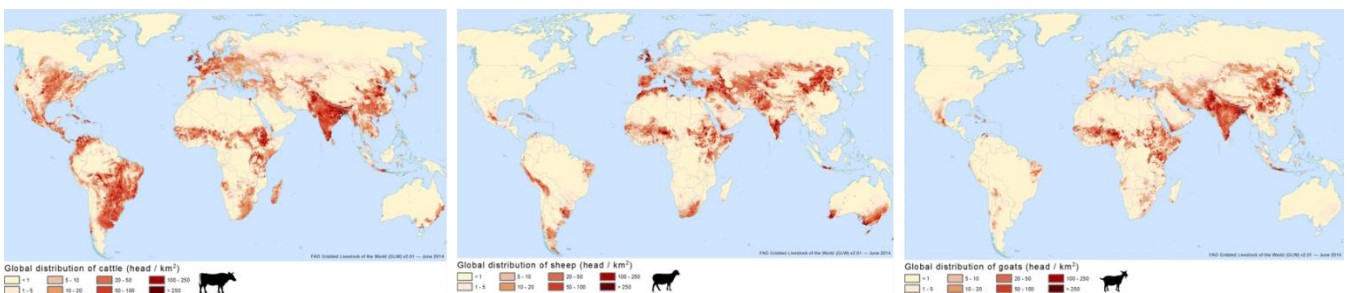


**Figure 3. Global distributions of cattle, sheep and goats, Gridded Livestock of the World (Robinson et al., 2014)**

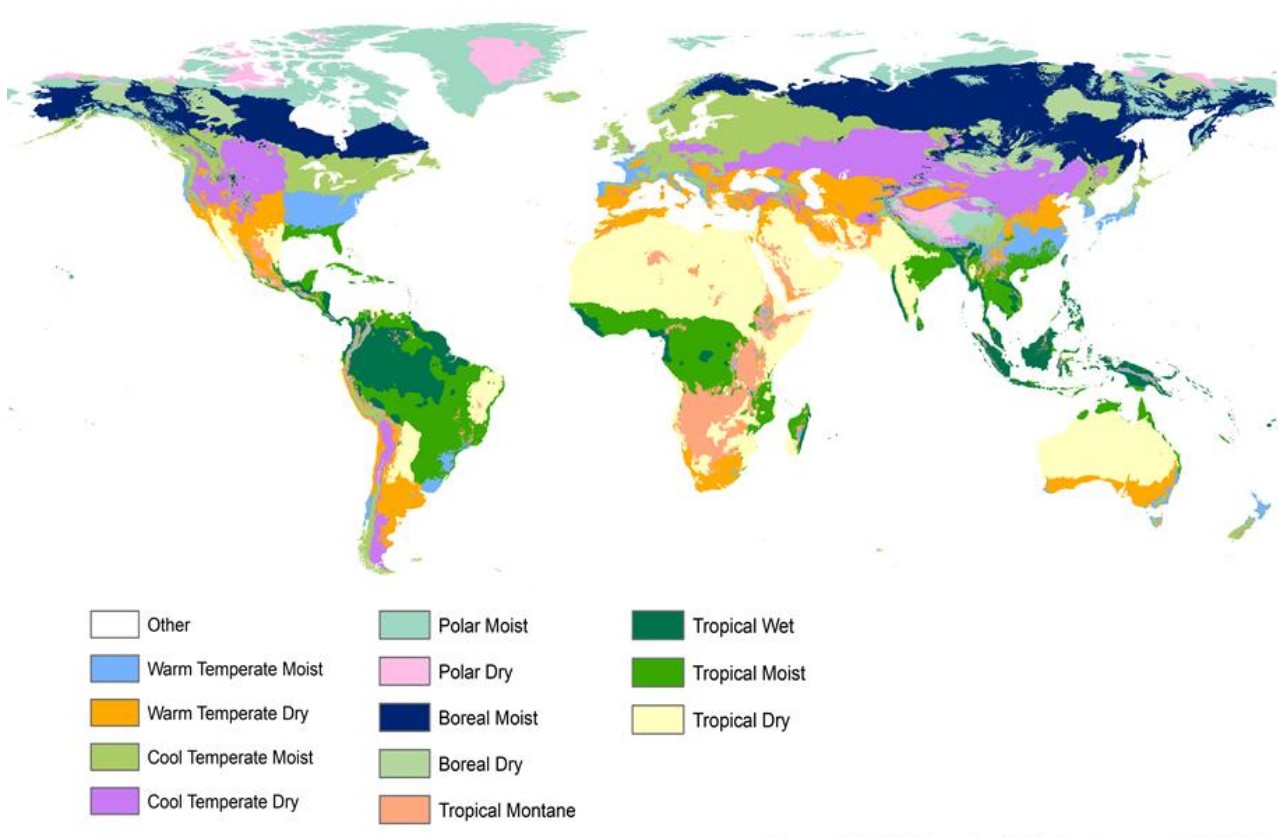


**Figure 4. Climatic zones based on IPCC classification (JRC, 2010)**




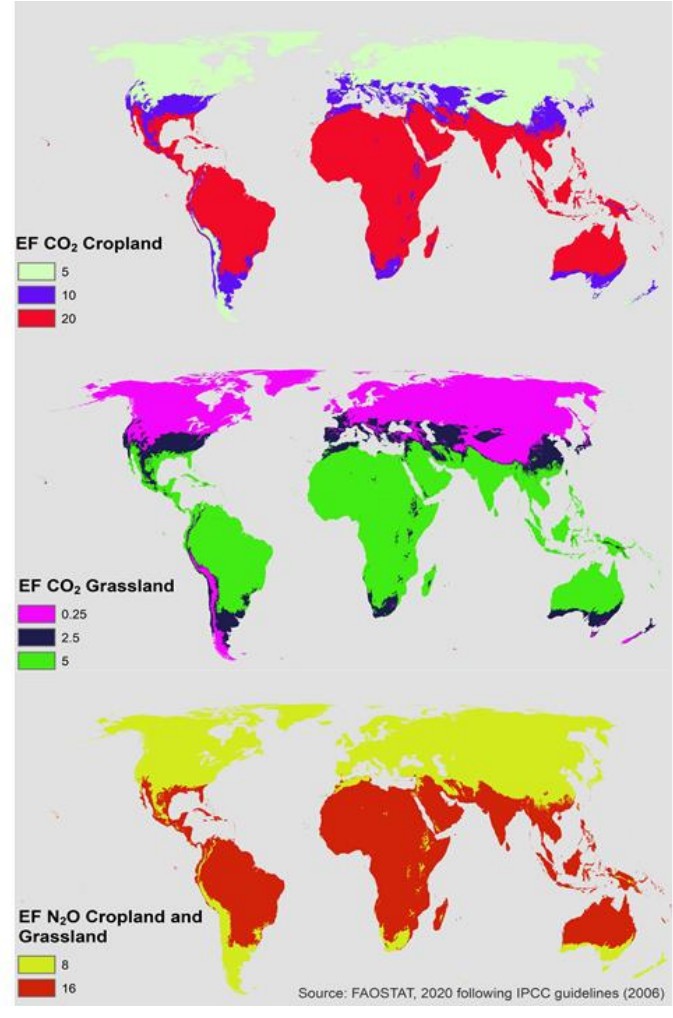

**Figure 5. Spatial layers of emission factors by gas, land use and climatic zone**

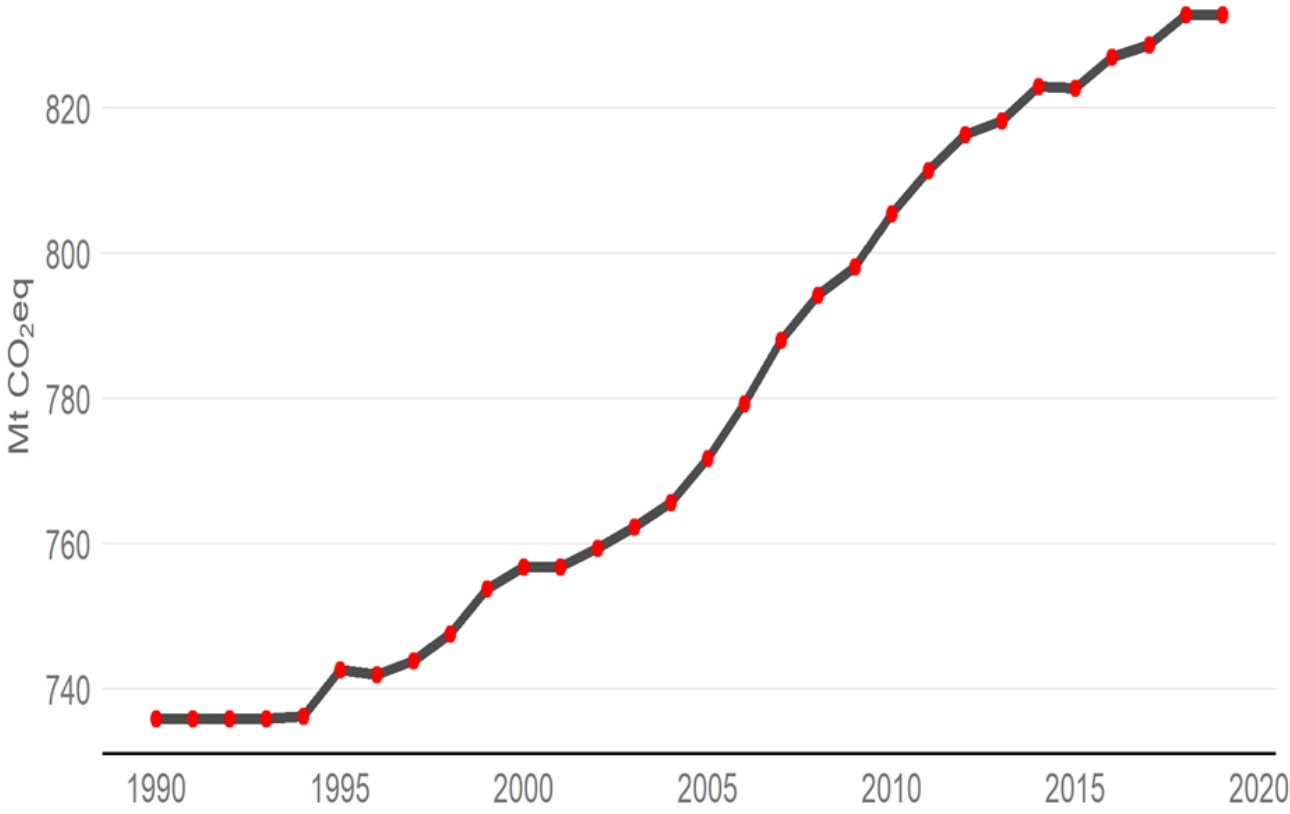


**Figure 6. Global emissions from drained organic soils, 1990–2019**


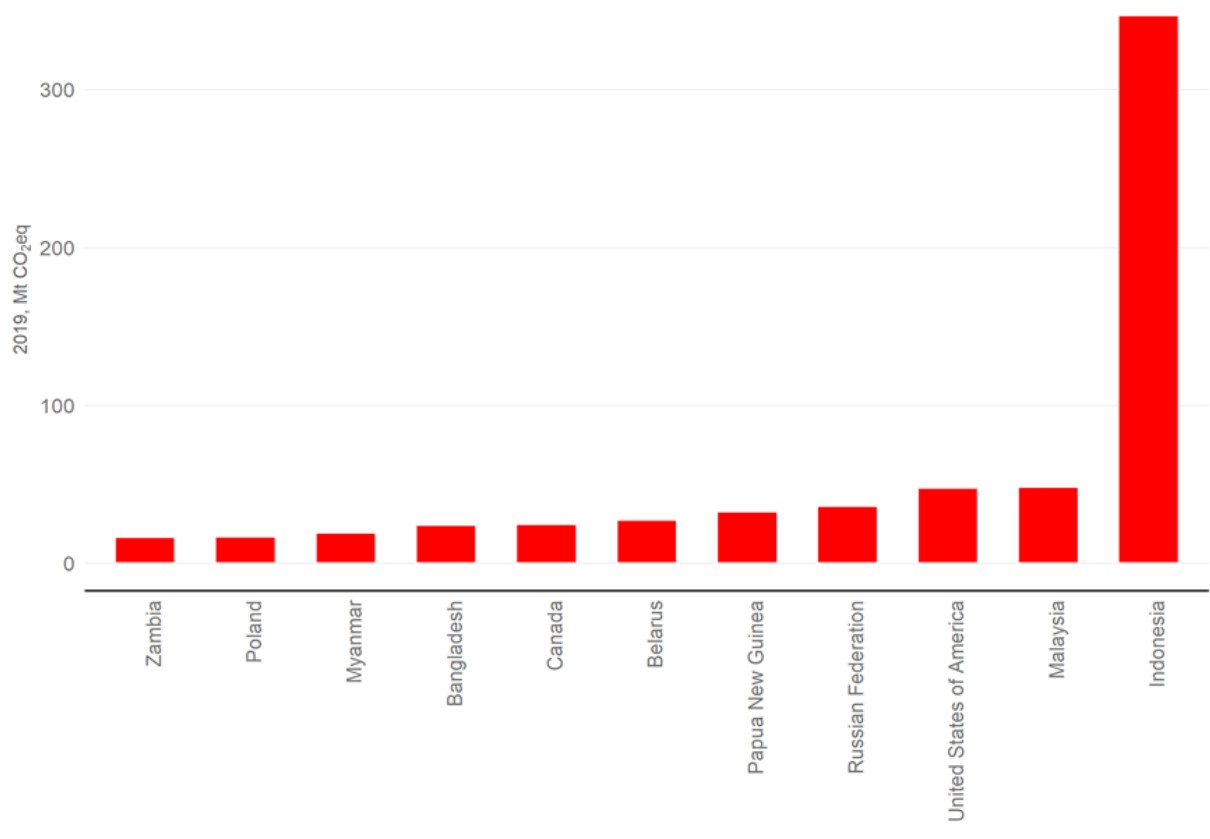

Source: FAOSTAT, 2020

**Figure 7. Top 10 countries by emissions from drained organic soils (75 percent of global emissions)**


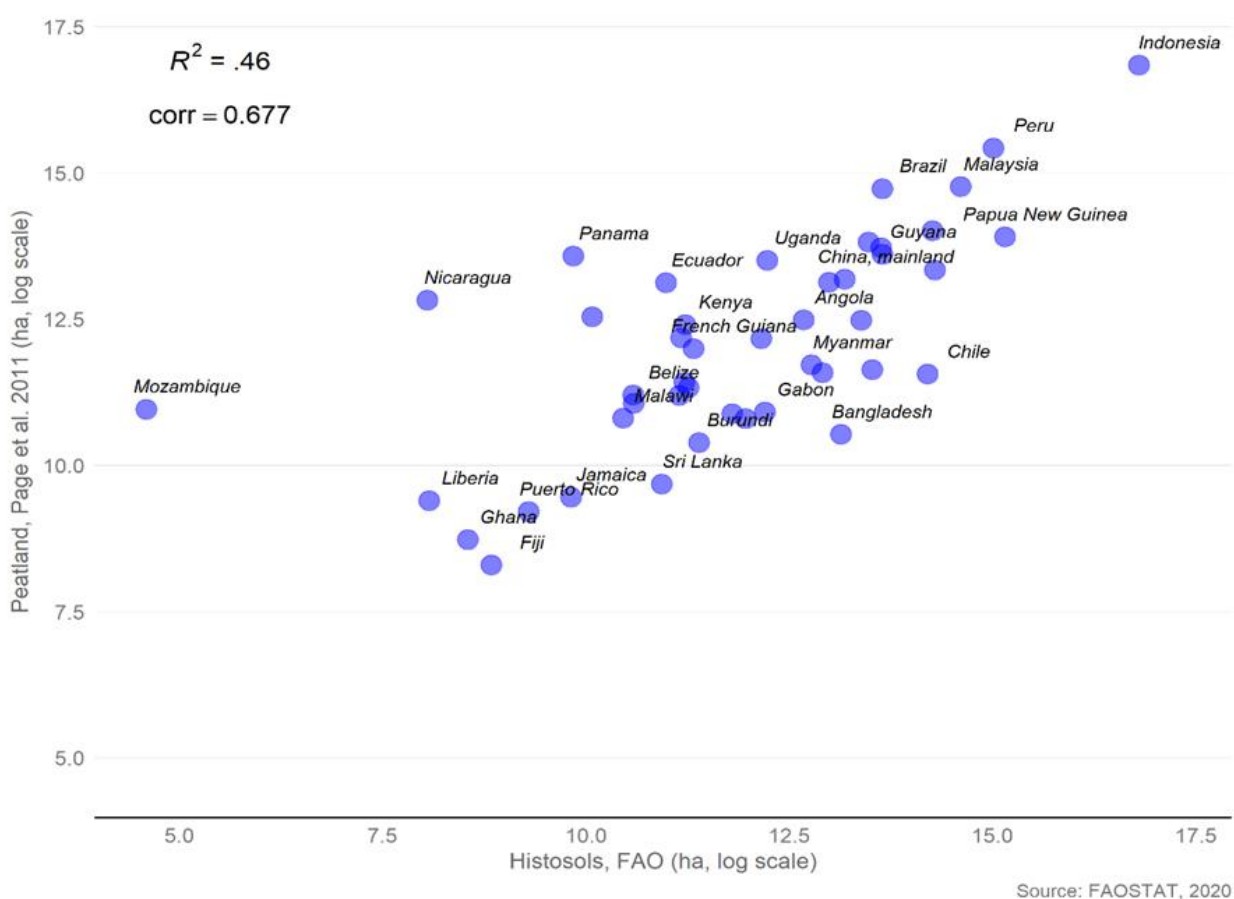

**Figure 8. Scatterplot of log-transformed area estimates for organic soils (Page et al., 2011) and FAOSTAT.**

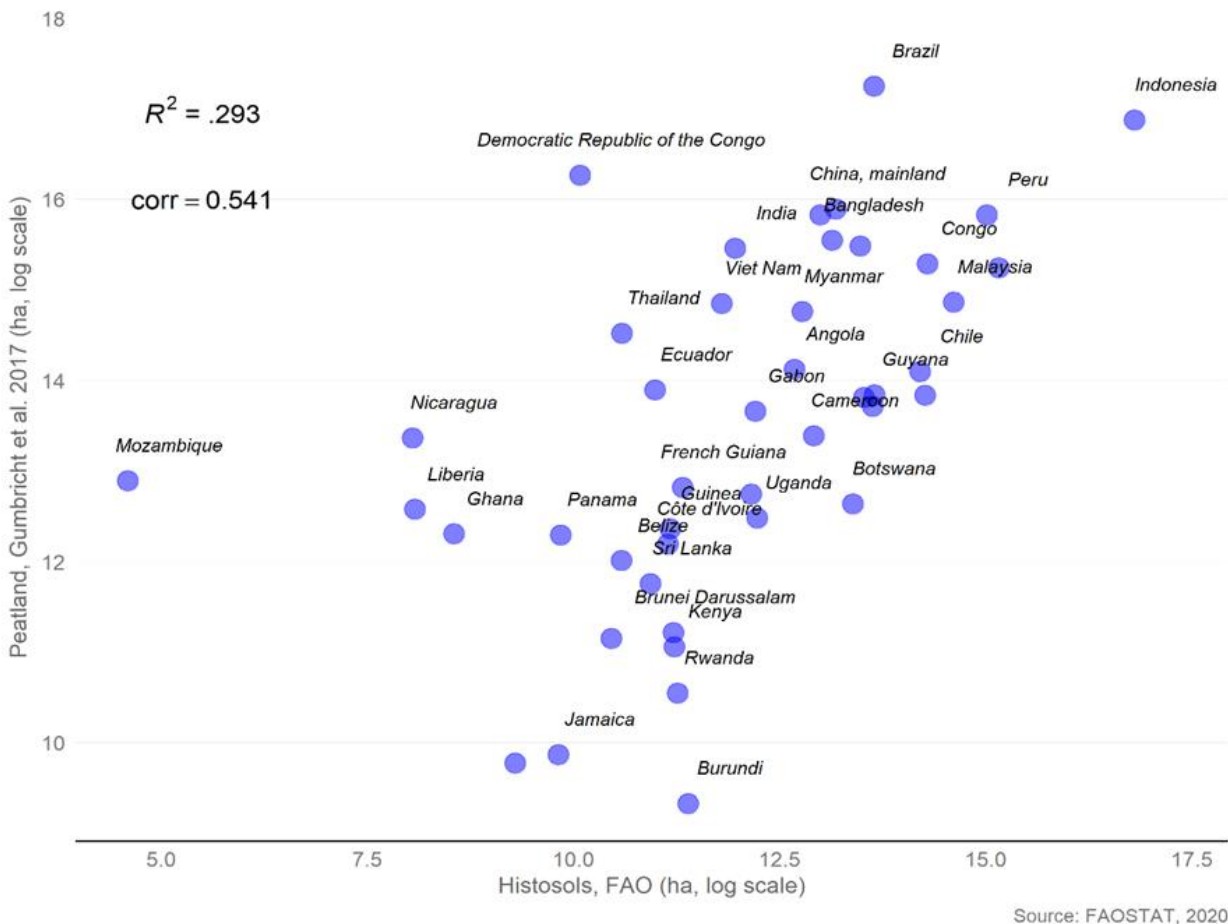


**Figure 9. Scatterplot of area estimates for organic soils in published data (Gumbricht et al., 2017) and FAOSTAT. Data have been log-transformed to avoid dependence on a few large vales.**


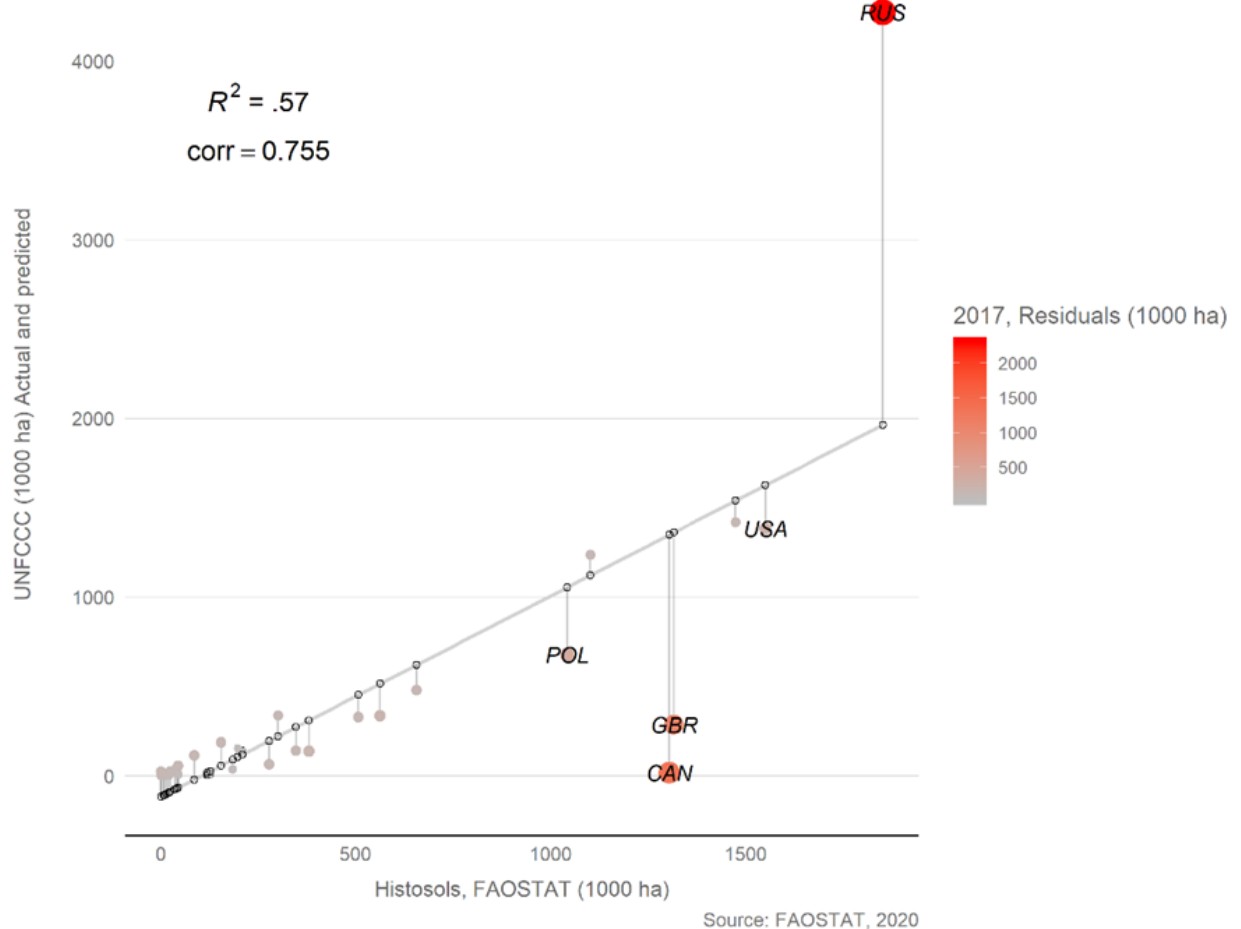

**Figure 10. Comparison of FAOSTAT estimates of drained organic soils area *vs* official country data reported to UNFCCC (year 2017). Distance from predicted (on the fitted line) and actual data**







**Figure 11. Comparison of FAOSTAT estimates of GHG emissions ($N_2O$) compared to UNFCCC data (year 2017)**


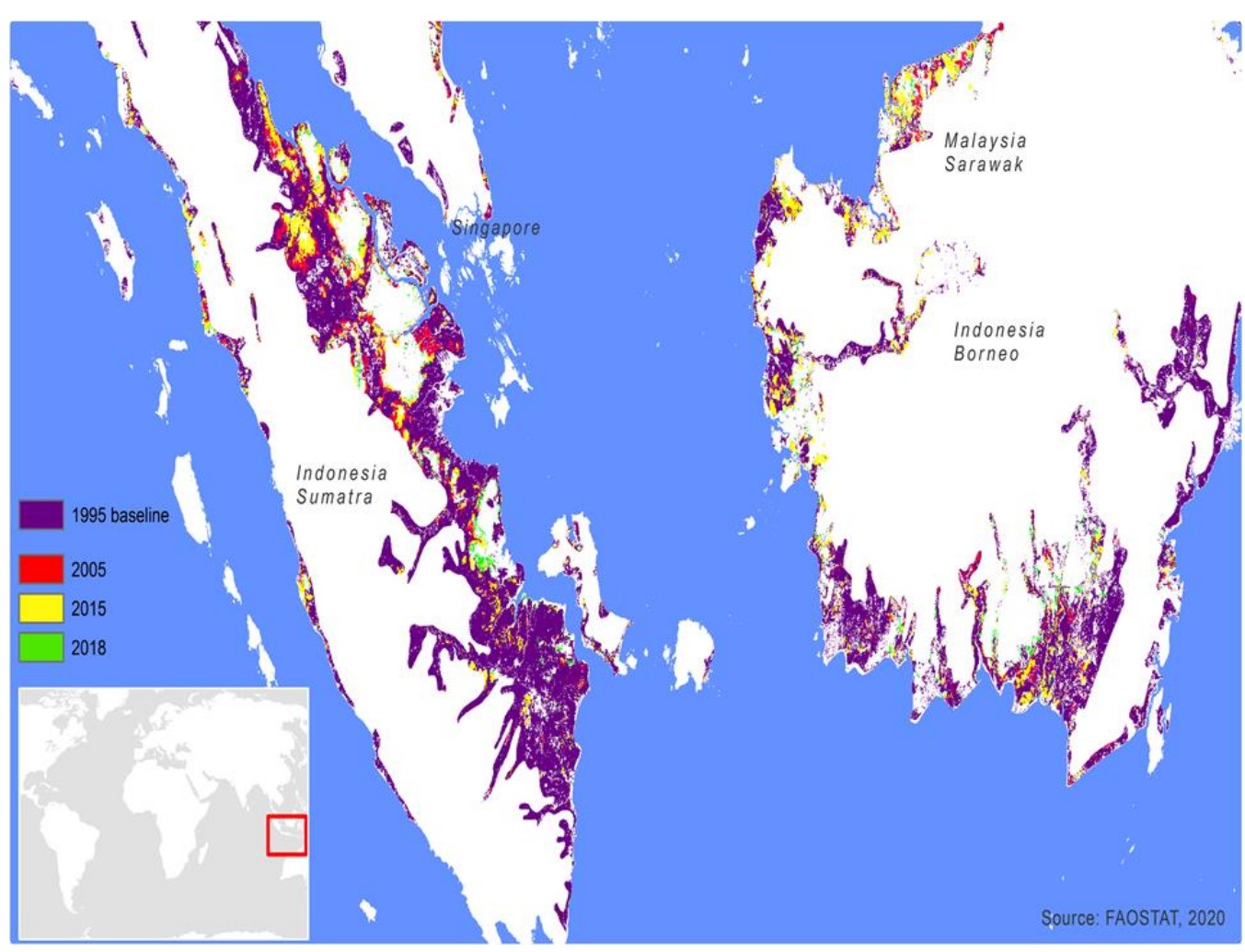


**Figure 12. FAOSTAT estimates of the extent of drained organic soils in Indonesia and Malaysia over time, showing total drained area in 1995 and successive additions by 2005, 2015 and 2018**




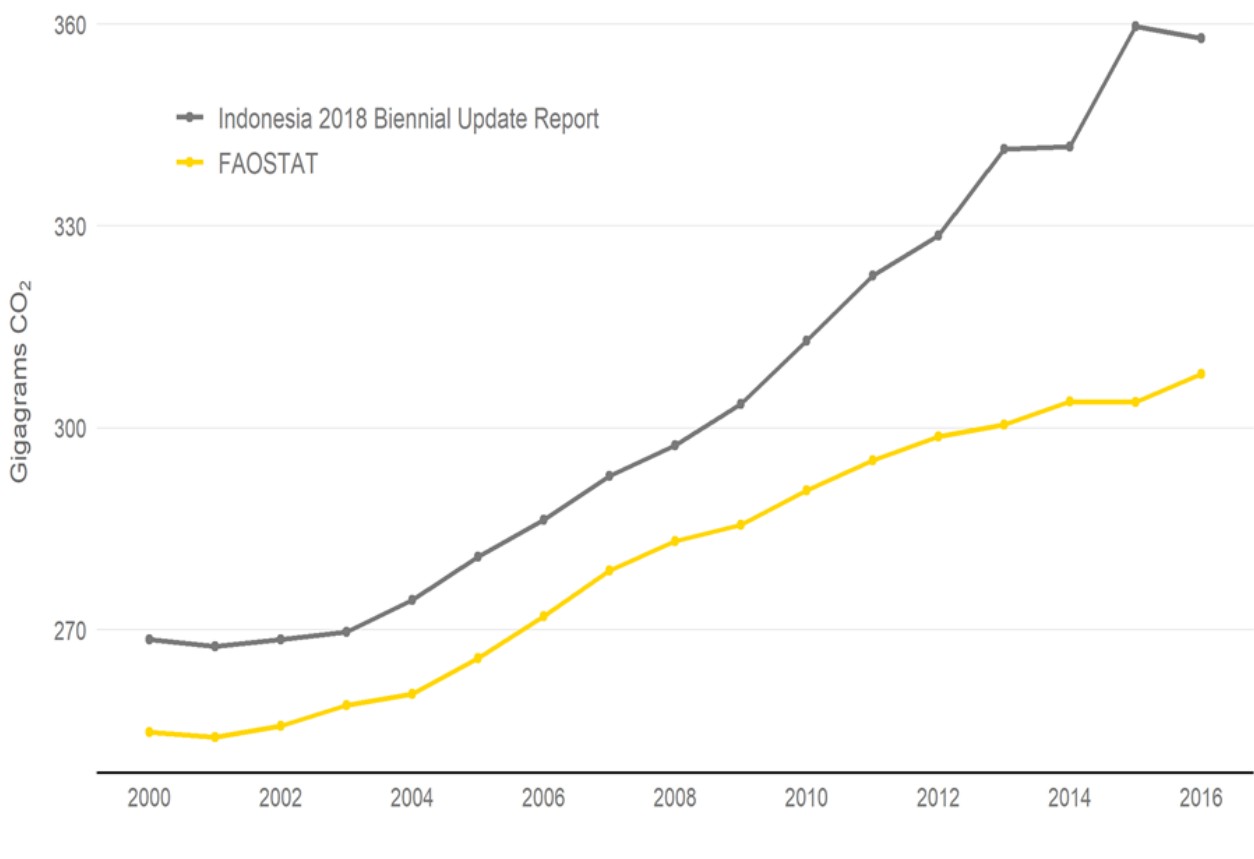

**Figure 13. FAOSTAT estimates *vs* reported CO₂ emissions form the drainage of organic soils in Indonesia, 1990–2016**




**Figure 14. Oil palm and other tree plantations (Petersen et al., 2016) and FAOSTAT Cropland organic soils in 2014**



## APPENDIX A: Additional results

Table A1. **Original area of** *histosols* **and shares of drained** *histosols* **in 1995[*] and 2019**

| Country | Area of histosols (ha) | *of which* drained (%) | |
|---|---|---|---|
| | | **1995** | **2019** |
| Turkey | 16 | 95.5% | 95.5% |
| Serbia | 90 | 0.0% | 92.7% |
| Guinea-Bissau | 101 | 14.2% | 14.2% |
| Luxembourg | 308 | 0.0% | 58.7% |
| Namibia | 853 | 15.0% | 15.0% |
| Solomon Islands | 1,062 | 1.0% | 1.0% |
| Isle of Man | 1,332 | 80.7% | 76.8% |
| Equatorial Guinea | 1,747 | 0.5% | 0.5% |
| Croatia | 2,987 | 12.6% | 11.3% |
| Nicaragua | 3,124 | 55.7% | 39.7% |
| Liberia | 3,208 | 56.4% | 74.8% |
| Eritrea | 3,485 | 2.3% | 2.3% |
| Slovakia | 4,294 | 58.4% | 56.0% |
| Albania | 4,509 | 85.9% | 84.0% |
| Ghana | 5,155 | 38.7% | 33.4% |
| Central African Republic | 5,745 | 12.2% | 17.3% |
| Fiji | 6,867 | 30.4% | 29.5% |
| Slovenia | 7,653 | 58.7% | 83.0% |
| Montenegro | 7,775 | 0.0% | 10.2% |
| Puerto Rico | 10,850 | 23.4% | 15.4% |
| Republic of Moldova | 12,274 | 52.7% | 47.2% |
| Bosnia and Herzegovina | 12,770 | 36.4% | 34.9% |
| Uruguay | 18,213 | 52.0% | 51.2% |
| Jamaica | 18,309 | 24.7% | 34.2% |
| Panama | 18,859 | 78.5% | 78.2% |
| Costa Rica | 21,135 | 18.5% | 14.7% |
| Belgium | 22,985 | 0.0% | 34.3% |
| Democratic Republic of the Congo | 23,750 | 36.8% | 35.7% |
| Portugal | 25,810 | 50.1% | 48.4% |
| Austria | 27,863 | 41.2% | 45.8% |
| Italy | 28,540 | 81.4% | 81.3% |
| South Africa | 31,955 | 54.3% | 63.7% |



| | | | |
|---|---|---|---|
| Malawi | 34,745 | 45.2% | 45.2% |
| Spain | 36,030 | 41.3% | 50.7% |
| Czechia | 37,943 | 29.6% | 28.0% |
| Faroe Islands | 38,952 | 34.6% | 34.6% |
| Belize | 39,354 | 13.3% | 14.4% |
| Thailand | 39,548 | 65.4% | 63.0% |
| Bulgaria | 52,362 | 76.6% | 73.8% |
| Greece | 55,569 | 81.8% | 79.8% |
| Sri Lanka | 55,942 | 57.6% | 54.4% |
| Ecuador | 58,961 | 1.0% | 3.3% |
| Côte d'Ivoire | 69,150 | 55.5% | 60.6% |
| Guinea | 71,016 | 27.0% | 26.3% |
| Brunei Darussalam | 73,964 | 9.0% | 7.5% |
| Kenya | 74,610 | 11.9% | 11.9% |
| Rwanda | 77,814 | 47.7% | 46.3% |
| French Guiana | 82,487 | 0.4% | 1.0% |
| Switzerland | 86,097 | 43.9% | 39.9% |
| Burundi | 88,387 | 77.0% | 78.8% |
| Denmark | 111,011 | 77.5% | 76.5% |
| Democratic People's Republic of Korea | 113,916 | 3.4% | 4.1% |
| Viet Nam | 132,725 | 52.0% | 49.9% |
| India | 156,362 | 67.0% | 65.5% |
| Madagascar | 189,666 | 55.4% | 57.6% |
| Gabon | 199,075 | 4.7% | 5.6% |
| Uganda | 204,211 | 57.1% | 56.7% |
| Nepal | 233,847 | 31.2% | 32.8% |
| Romania | 248,517 | 7.7% | 7.7% |
| New Zealand | 254,339 | 50.2% | 49.8% |
| Hungary | 275,678 | 71.2% | 69.0% |
| Ethiopia | 289,128 | 44.1% | 44.9% |
| France | 308,893 | 70.7% | 67.8% |
| Angola | 319,617 | 2.1% | 2.1% |
| Myanmar | 352,812 | 83.0% | 81.8% |
| Japan | 358,961 | 52.4% | 42.9% |
| Netherlands | 395,113 | 78.4% | 75.9% |
| Cameroon | 404,266 | 6.0% | 7.0% |
| Colombia | 437,958 | 2.8% | 4.0% |



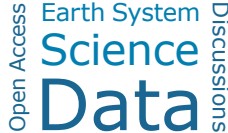

| | | | |
|---|---:|---:|---:|
| Australia | 440,351 | 27.0% | 27.1% |
| United Republic of Tanzania | 492,667 | 21.8% | 20.6% |
| Bangladesh | 507,083 | 69.4% | 66.9% |
| China, mainland | 530,701 | 27.5% | 28.2% |
| Botswana | 651,384 | 1.5% | 1.7% |
| Falkland Islands (Malvinas) | 667,141 | 43.3% | 43.2% |
| Iceland | 684,893 | 6.4% | 6.4% |
| Argentina | 694,519 | 29.4% | 30.1% |
| Lithuania | 701,767 | 51.4% | 49.4% |
| Venezuela (Bolivarian Republic of) | 710,571 | 4.0% | 5.7% |
| Latvia | 735,751 | 23.3% | 26.9% |
| Suriname | 746,249 | 1.0% | 1.7% |
| South Sudan | 827,363 | 0.0% | 32.9% |
| Brazil | 840,917 | 1.3% | 1.7% |
| Guyana | 844,866 | 7.4% | 9.4% |
| Estonia | 918,164 | 16.4% | 20.1% |
| Ireland | 1,118,046 | 51.8% | 50.3% |
| Ukraine | 1,262,568 | 55.9% | 52.0% |
| Mongolia | 1,311,509 | 80.6% | 81.0% |
| Chile | 1,472,126 | 3.3% | 3.4% |
| Germany | 1,482,858 | 76.0% | 74.4% |
| Zambia | 1,565,696 | 23.3% | 23.2% |
| Congo | 1,609,628 | 3.3% | 3.5% |
| Poland | 1,769,225 | 61.0% | 59.0% |
| Norway | 1,947,518 | 13.1% | 14.4% |
| Malaysia | 2,210,193 | 20.3% | 30.4% |
| United Kingdom | 2,610,052 | 51.4% | 50.4% |
| Belarus | 3,014,298 | 49.7% | 48.9% |
| Peru | 3,300,367 | 0.0% | 0.1% |
| Papua New Guinea | 3,806,847 | 10.6% | 11.5% |
| Sweden | 6,797,032 | 4.4% | 6.0% |
| Finland | 9,205,429 | 4.5% | 5.7% |
| Indonesia | 19,791,043 | 19.9% | 24.4% |
| United States of America | 25,399,312 | 6.2% | 6.1% |
| Canada | 105,758,515 | 1.2% | 1.2% |
| Russian Federation | 116,116,633 | 1.6% | 1.6% |





| World | 328,935,932 | 7.0% | 7.5% |
|---|---|---|---|

[a] 1995 is chosen arbitrarily to account for the reporting of countries after the split of the Soviet Union.




Table A2. **Global emissions in 1990 and 2019 by gas and by land use**

| Land use | Mt by gas | | Total in Mt CO₂eq |
|---|---|---|---|
| | N₂Oª | CO₂ | |
| **Cropland organic soils** | 66.4 | 589.9 | 656.3 |
| **Grassland organic soils** | 34.9 | 44.8 | 79.6 |
| Total emissions in 1990 | **101.2** | **634.7** | **735.9** |
| | | | |
| **Cropland organic soils** | 74.4 | 675.9 | 750.3 |
| **Grassland organic soils** | 36.0 | 46.5 | 82.5 |
| Total emissions in 2019 | **110.5** | **722.4** | **832.9** |

ª N₂O emissions converted to CO₂eq through IPCC AR5 GWP (IPCC, 2014b).





**APPENDIX B: Tables for validation**

Table B1. **Peat extent from Page et al., 2011, Gumbricht et al., 2017, and FAO area of *histosols*, all in 1000 ha**

|  |  | Page et al., 2011 | Histosols (FAO) | Gumbricht et al. 2017 |
|---|---|---|---|---|
|  |  | Best estimate from meta-analysis | Spatial layers | |
| **Africa** | Angola | 264 | 320 | 1,359 |
|  | Botswana | 265 | 651 | 308 |
|  | Burundi | 33 | 88 | 11 |
|  | Cameroon | 108 | 404 | 654 |
|  | Congo | 622 | 1,610 | 4,357 |
|  | Democratic Republic of the Congo | 280 | 24 | 11,592 |
|  | Gabon | 55 | 199 | 855 |
|  | Ghana | 6 | 5 | 221 |
|  | Guinea | 195 | 71 | 234 |
|  | Côte d'Ivoire | 73 | 69 | 198 |
|  | Kenya | 244 | 75 | 64 |
|  | Liberia | 12 | 3 | 290 |
|  | Madagascar | 192 | 190 | 343 |
|  | Malawi | 49 | 35 | 70 |
|  | Mauritania | 6 | 0 | 0 |
|  | Mauritius | 0 | 0 | 0 |
|  | Mozambique | 58 | 0 | 398 |
|  | Nigeria | 184 | 0 | 0 |
|  | Réunion | 0 | 0 | 0 |
|  | Rwanda | 83 | 78 | 38 |
|  | Senegal | 4 | 0 | 0 |
|  | Sierra Leone | 0 | 0 | 0 |
|  | South Sudan | 907 | 827 | 906 |
|  | Uganda | 730 | 204 | 263 |
|  | Zambia | 1,220 | 1,566 | 1,021 |
|  | **Africa *total*** | **5,586** | **6,419** | **23,182** |
| **Asia (South East)** | Brunei Darussalam | 91 | 74 | 74 |
|  | Indonesia | 20,695 | 19,791 | 21,342 |
|  | Malaysia | 2,589 | 2,210 | 2,858 |
|  | Myanmar | 123 | 353 | 2,577 |
|  | Papua New Guinea | 1,099 | 3,807 | 4,163 |



| | | | | |
|---|---|---|---|---|
| | Philippines | 65 | 0 | 0 |
| | Thailand | 64 | 40 | 2,024 |
| | Viet Nam | 53 | 133 | 2,815 |
| | **Asia (South East)** *total* | **24,778** | **26,407** | **35,851** |
| **Asia (other)** | Bangladesh | 38 | 507 | 5,667 |
| | China | 531 | 531 | 8,001 |
| | India | 49 | 156 | 5,179 |
| | Sri Lanka | 16 | 56 | 128 |
| | **Asia (other)** *total* | **634** | **1,250** | **18,974** |
| **Central America & Caribbean** | Belize | 74 | 39 | 165 |
| | Cuba | 364 | 0 | 0 |
| | El Salvador | 9 | 0 | 0 |
| | Haiti | 119 | 0 | 0 |
| | Honduras | 453 | 0 | 0 |
| | Jamaica | 13 | 18 | 19 |
| | Mexico | 100 | 0 | 0 |
| | Nicaragua | 371 | 3 | 638 |
| | Panama | 787 | 19 | 218 |
| | Puerto Rico | 10 | 11 | 18 |
| | Trinidad and Tobago | 1 | 0 | 0 |
| | **Central America & Caribbean** *total* | **2,300** | **90** | **1,058** |
| **Pacific** | Australia | 15[a] | 440 | 2,142 |
| | Fiji | 4 | 7 | 0 |
| | **Pacific** *total* | **19** | **447** | **2,142** |
| **South America** | Bolivia (Plurinational State of) | 51 | 0 | 0 |
| | Brazil | 2,500 | 841 | 30,965 |
| | Chile | 105 | 1,472 | 1,327 |
| | Colombia | 504 | 438 | 7,473 |
| | Ecuador | 500 | 59 | 1,084 |
| | French Guiana | 162 | 82 | 370 |
| | Guyana | 814 | 845 | 1,028 |
| | Peru | 5,000 | 3,300 | 7,499 |
| | Suriname | 113 | 746 | 998 |
| | Venezuela (Bolivarian Republic of) | 1,000 | 711 | 5,300 |
| | **South America** *total* | **10,749** | **8,495** | **56,045** |
| | | | | |
| **Total** | | **44,066** | **43,108** | **137,252** |

[a] In Page et al., 2011, Australia estimates limited to Australia, Queensland.






Table B2. **FAOSTAT estimates and UNFCCC reported country data: area drained and N₂O (kt) emissions, by country in Annex I group[a], 2017**

| ISO3 | Country | UNFCCC | FAOSTAT | UNFCCC | FAOSTAT |
|------|---------|--------|---------|--------|---------|
| | | *Area drained (ha)* | | *N₂O emissions (kt)* | |
| AUS | Australia | 4,000 | 119,195 | 0.05 | 1.51 |
| AUT | Austria | 12,954 | 12,763 | 0.17 | 0.16 |
| BEL | Belgium | 2,520 | 7,899 | 0.03 | 0.10 |
| BGR | Bulgaria | 41,267 | 38,750 | 0.52 | 0.49 |
| BLR | Belarus | 1,419,100 | 1,474,262 | 17.84 | 18.53 |
| CAN | Canada | 16,156 | 1,304,454 | 0.20 | 16.27 |
| CHE | Switzerland | 17,339 | 34,369 | 0.22 | 0.43 |
| CZE | Czechia | -- | 10,593 | -- | 0.13 |
| DEU | Germany | 1,235,057 | 1,102,052 | 9.52 | 13.77 |
| DNK | Denmark | 112,792 | 84,980 | 1.60 | 1.06 |
| ESP | Spain | -- | 18,342 | -- | 0.23 |
| EST | Estonia | 34,815 | 183,505 | 0.44 | 2.30 |
| FIN | Finland | 327,616 | 506,840 | 5.03 | 6.34 |
| FRA | France | 139,056 | 209,149 | 1.75 | 2.62 |
| GBR | United Kingdom | 285,700 | 1,316,388 | 3.6 | 15.9 |
| GRC | Greece | 6,665 | 44,520 | 0.08 | 0.56 |
| HRV | Croatia | 2,685 | 336 | 0.03 | 0.00 |
| HUN | Hungary | -- | 190,462 | -- | 2.39 |
| IRL | Ireland | 333,853 | 562,872 | 2.26 | 6.83 |
| ISL | Iceland | 55,598 | 43,859 | 0.08 | 0.51 |
| ITA | Italy | 25,480 | 23,247 | 0.32 | 0.29 |
| JPN | Japan | 185,592 | 154,160 | 0.39 | 1.89 |
| LTU | Lithuania | 138,758 | 346,350 | 1.74 | 4.35 |
| LUX | Luxembourg | -- | 181 | -- | 0.00 |
| LVA | Latvia | 152,160 | 197,363 | 2.71 | 2.48 |
| NLD | Netherlands | 337,102 | 300,076 | 2.36 | 3.77 |
| NOR | Norway | 63,862 | 277,520 | 1.26 | 3.36 |
| NZL | New Zealand | 8,020 | 126,770 | 0.10 | 1.51 |
| POL | Poland | 678,000 | 1,042,266 | 8.52 | 13.08 |
| PRT | Portugal | -- | 12,598 | -- | 0.15 |



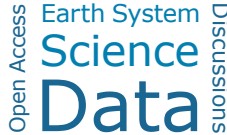

| | | | | | |
|---|---|---|---|---|---|
| ROU | Romania | 6,387 | 19,234 | 0.08 | 0.24 |
| RUS | Russian Federation | 4,274,300 | 1,852,512 | 53.93 | 23.24 |
| SVK | Slovakia | -- | 2,399 | -- | 0.03 |
| SVN | Slovenia | 2,501 | 6,361 | 0.03 | 0.08 |
| SWE | Sweden | 136,692 | 379,122 | 2.79 | 4.71 |
| TUR | Turkey | 21,840 | 15 | 0.27 | 0.00 |
| UKR | Ukraine | 478,400 | 656,586 | 6.01 | 8.25 |
| USA | United States of America | 1,383,162 | 1,551,534 | 19.56 | 26.48 |
| **Total** | | **11,939,429** | **14,213,882** | **144** | **184** |

[a] Data for this reporting category are not occurring (NO) in the UNFCCC tables for Cyprus, Czechia, Hungary, Kazakhstan, Luxembourg, Malta, Monaco, Portugal and Spain. Data were not estimated in Slovakia (NE).