# Peer review of "Drainage of organic soils and GHG emissions: Validation with country data"

_Earth System Science Data, 2020_

## Referee Comment (RC1) · Anonymous Referee #1 · 28 Aug 2020

This study presents the approach and main results of a new methodology developed for FAOSTAT. By combining overlays of maps of land cover with the distribution of wetland soils (histosols) and IPCC emission factors, the authors present a global annual dataset of peatland drained area and greenhouse gas emissions ($CO_2$ and $N_2O$) over a time series encompassing three decades (1990–2019). This allows the authors to identify trends in drained areas and emissions over time and to validate the FAO emissions estimates with country data. Sources of uncertainty are discussed. Importantly, the FAO dataset currently provides the only available country/regional/global time series data on GHG emissions from drained organic soils, thereby supporting analysis of trends and the identification of current or emerging emissions hotspots that could be targeted for mitigation measures. The paper is generally well written with a clear

description of methodological approach, limitations and uncertainties, although I do have some suggestions for further improvement of several aspects relating to uncertainty. The results are very relevant to current actions to reduce land-use derived GHG emissions; they are generally well presented and discussed. I recommend publication following minor revision – see my specific comments and suggestions below. Specific comments: Line 25 – change wet soils ecosystems to wet soil ecosystems Line 46 – by citing Rieley & Page (2016) you are only referring to tropical peatlands – please include an additional balancing reference for northern peatlands Section 2 – can the authors acknowledge that by using data on the distribution of histosols as a proxy for peat soils, some areas of histosols will be included that are not strictly defined as peat soils (e.g. if one followed the definition of a minimum peat depth of 40 cm with organic content > 65%) Line 105 – suggest rephrase: In order to support crop cultivation activities, organic soils need to be drained Lines 106 – 107 – sentence on livestock needs to be rephrased – sense is not clear : grazing per se does not result in drainage Lines 115-116 – what are the range of values for soil carbon content, pH, water storage content used to characterise histosols? Line 122 – replace Spatial with Space (European Space Agency) Line 147 – replace climatic zones with climate zone Line 164 – remove 'and' Line 165 – section 2.6 Limitations and uncertainty – a) Would the authors consider applying and including emissions based on the revised IPCC emission factors presented in the updated 2013 IPCC guidelines? Perhaps presented alongside the EFs from the 2006 guidelines? For the most part, the 2013 EFs are based on a wider literature base and provide a more accurate assessment of Tier 1 emissions across land-use categories/climate zones. Alternatively, the authors should at least acknowledge and discuss how use of the 2013 EFs would alter their emissions estimates. b) Can the authors consider adding a further couple of sentences into this section on the uncertainties that arise, over time, from peat wastage – i.e. where drainage leads to the depletion and eventual loss of organic matter from shallow peat soils there is the potential for a change in the scale of emissions. Without accurate country data on peat depth and rate of peat loss it will not be possible to estimate peat depletion rates, but
this could at least be acknowledged. N.B. At least in drained temperate peatlands, a reduction in soil organic carbon does not necessarily result in a reduction in CO2 emissions (e.g. see Tiemeyer et al. 2016 - https://doi.org/10.1111/gcb.13303), but in tropical peatlands peat loss is usually accompanied by an increasing occurrence of flooding which will necessarily reduce CO2 emissions over time. Line 187 – the authors could considering clarifying here, or in the discussion, that whilst the analysis is only for drained peat soils under cropland and grassland, in fact in some countries (e.g. Indonesia) there are extensive additional areas of peatland subject to drainage that are under other land covers (e.g. degraded forest, scrub in the case of countries in SE Asia) and emissions from these land covers are not captured in this analysis. Line 211 – add a full stop after ranges; change estimates to estimate. Line 218 and following – there is indeed a discrepancy between estimates in Page et al. (2011) and the data presented by Gumbricht et al. (2017), particularly in relation to S. America. The authors might wish to expand here on why these discrepancies could have arisen e.g. the remote sensing approach (remote sensed wetness index) used by Gumbricht provides very limited data over tropical forested peatlands and therefore in these areas their estimates appear to be more based on topography, climatic wetness etc – which may be reasonable assumptions for predicting the location of wetlands but cannot be used to determine whether or not these wetlands are peat forming systems. The estimates for Brazil likely therefore indicate extensive areas of wetland, but not necessarily peatland. Line 224 – replace 'both about a third' with 'but both estimates are about a third of …..' Line 232 – change explains to explain Line 233 – change 'For one percent' to 'For a one percent . . .' Line 239 – change consistently to consistent Line 242 – change peatlands to peatland Line 243/section 4.1 – I would encourage the authors to also mention that their estimates of emissions do not, for example, include emissions from water surfaces (e.g. CO2/CH4 evasion from drainage channels, e.g. in plantation landscapes). Nor do they include fire emissions. In SE Asia, GHG emissions from peat fires can be of a comparable magnitude to emissions arising from peat oxidation driven by drainage and agricultural uses. But peat fires are also an increasing feature

of other drained peatlands – e.g. in Russia. Line 261 – change organic area to organic soil area Line 270 – change 'due to' to 'be due to' Line 278 – country name is missing before the final bracketed numbers. The difference here in emissions seems particularly large (16 vs. 0.2 kt N20) – do the authors have an explanation for this? Line 285 – can the authors provide some more detail on why the Tiemeyer emissions estimates for organic soils in Germany are so much higher than FAOSTAT emissions? Line 288 – replace fourty with forty Section 4.2.2. – at the start of this section you refer to both Indonesia and Malaysia, but then go on to only compare the FAOSTAT and country data on emissions for Indonesia. For completeness, is it possible to also include a comparison of the Malaysian datasets? Also, Miettinen et al (2016) give the area of peatland under crops (plantations and smallholder agriculture) in Indonesia as 6.3 Mha compared with the FAOSTAT estimate of 5 Mha. Perhaps worth mentioning this difference. Does the 5 Mha area estimated in FAOSTAT include all plantations (including pulpwood) or only oil palm and other food crop plantations? Your Table 6 implies you include all types of plantations (but this should be clarified). Line 302 – insert 'a' before 'main driver' Line 303 – Hooijer is mis-spelt (2010 citation) Lines 317 – 318 – improve expression – sense not clear Line 320 – insert 'be' after 'may' Line 335 – should 'disseminated' read 'disaggregated' ? Sense not clear Line 352 – insert 'to' before 'whether' Table 6 – please clarify whether the 'all plantations' category includes pulpwood plantations as well as oil palm and other food crop plantations (e.g. coconut) (see point above on Section 4.2.2). Table 7 – correct mis-spelling of Hooijer. Also, some of these studies (e.g. Hooijer et al. 2012, Cooper et al. 2020) take account of the initial pulse of carbon that is lost from peat soils in the immediate (up to 5) years following peatland drainage and deforestation. Other studies, however, do not account for this initial pulse and represent emissions once the peat landscape has stabilised under the new land use (+5 years after drainage). Figure 5 – the figure caption should indicate that the emission factors are derived from IPCC (2006). Figure 13 – in the relevant part of the discussion, the discrepancy in the FAOSTAT estimate of emissions and the country reported emissions in Indonesia should be addressed. Could the discrepancy (lower FAOSTAT estimate) but due to the in-country data reporting emissions from all forms of degraded peatland land covers/uses, i.e. not just cropland/grassland? For example, the INCAS (Indonesian Carbon Accounting System) reports emissions from degraded, non-agricultural peatland (e.g. degraded forest and scrub).

---

## Referee Comment (RC2) · Anonymous Referee #2 · 8 Sep 2020

This study is highly valuable and timely in the context of GHG mitigation strategies and country submissions and reporting under UNFCCC and commitments to the Paris Agreement. Currently FAO is the only global consistent database providing information on activity data, emission factors and GHG emissions from drained organic soils, and not only. The authors update the old static map of drained organic soils from the year 2000 and their $CO_2$/$N_2O$ emissions with a new methodology developed for FAO-STAT which includes dynamic maps. The authors present times series of global annual dataset of drained area and $CO_2$/$N_2O$ emissions between 1990 - 2019 and validate it with country information. Some uncertainty information is provided but would be very useful if uncertainties on emissions could be quantified. I would also encourage in the future updates, the use of more recent land use and land use change products

(e.g. HILDA+) (https://landchange.imk-ifu.kit.edu/news/sneak-preview-hilda-coming ).
Paper reads well but I would suggest to be read by a native speaker to help improving
its flow. Similar to reviewer #1, I agree to its publication after minor/technical revisions.

Specific technical comments: Line 31: . . .large quantities of available organic sub-
strate. Line 32: replace "and especially since 1990" with "the world, especially after the
90s. . ." and perhaps add in brackets where oil palm became permanent crop Line 45:
to be clear if wetland condition or wet condition. Agree with referee #1: add references
not only for boreal but also for alpine organic soils (bogs, fens) Line 48: delete indeed
Line 56: as "as they continue emitting.." Lines 74-76: I would reference or name in
brackets all the maps used in this study (land use, density etc.) when they appear for
the first time Line 105: delete "indeed be" Line 139: how about other species? I guess
for the boreal areas with organic soils other animals are present – e.g. reindeers? Line
144: which map did you use from the JRC? Please reference/name the original map
as well. Line 154: please reference the Climate Convention Line 167: you mention
here drained peats: is it only peat or drained organic soils in general? Line 178: "data
suggests" Line 182: To which period are you referring to about Asia drainage (30%)?
Over the whole studied period or one particular year? Line 191: If 833 Mt refers here to
2019 then I would reformulate: "In 2019, global GHG emissions from drained organic
soils were 833 Mt CO2eq. They were 13 % and 10 % higher when compared to 1990
and 2000 respectively, representing 8 % . . .. " Line 194 and 195: I would delete gas.
Are the global emissions/all emissions you refer here total GHG emissions in CO2 or
CO2eq or total emissions from drainage? Please explain. Lines 211-215: I would add
to table 5 all specific comparisons. I was also wondering why you are using the old ref-
erence of Joosten 2002 and not a more recent updated information from his peatland
database which I think Prof. Joosten is updating regularly for the areas and emissions
from organic soils. (https://greifswaldmoor.de/global-peatland-database-en.html) Line
227: countries from South and Central America Lines 235-240: do you know what
causes the main difference between the way Gumbricht, Page and FAOSTAT calculate
country level estimates? Are these uncertainties due to area, method, level of detail

or input to the maps? Perhaps add a sentence at the end of the paragraph summarizing these differences. Line 247: please specify which UNFCCC data was used? 2019, 2020? Same for line 252: UNFCCC (year) data are available... Lines 262: do you know why these differences? I think it should be mentioned that Canada uses a high Tier model (CBM) to report to the UNFCCC. Line 266: I would name LULUCF sector and not category. As you define further, categories are 4.B, 4.C etc. Line 270: higher Tiers than... Line 271: delete As Line 282: please reference the IPCC Wetlands EFs, are the values from the Wetlands Supplement or the IPCC 2006 chapter 7? Line 295: delete the in "vs the 304". Please add everywhere the year for the UNFCCC data. Line 309: to those from established or better peer-reviewed literature Line 311: please check references: Petersen or Peterson Line 318: which emissions (CO2, N2O, total?) in this country? Emissions were due to.. Please add a % in brackets. Line 320: may less....please complete: may be less or may not be less important... Line 322: are direct measurements the in-situ measurements? And typically analyse.. Line 336: available Line 339: million tonnes, be consistent until now Mt was used Line 351: consistent with writing IPCC Figures 11 and 13: why the use of both gigagrams and kt?

---

## Referee Comment (RC3) · Anonymous Referee #3 · 24 Sep 2020

This study is a highly valuable and useful further development of the dataset on drained organic soils emissions already available on FAOSTAT. While that dataset was a picture of the global situation in 2000, here the authors produced a spatially-explicit timeseries of global estimations for the period 1990-2019 through the use of the ESA CCI Land Cover dataset, which offers yearly global land cover maps for the years 1992-2018. The other datasets remained those used in producing the previous FAOSTAT dataset: the HWSD map to identify histosols (adopted as proxy for organic soil, based on the IPCC guidelines), and the FAO Gridded Livestock of the World to identify grazed land. The work is extremely valuable due to the importance of emissions from drained organic soils in the global carbon budget, and the very limited data available about this carbon pool and GHG source. Organic soils contain about 30% of the total soil carbon

despite their relatively limited global area, and drainage of organic soils for agriculture or grazing purposes releases enormous amounts of carbon and N2O (a very powerful GHG) for long time periods (decades after the drainage). Drainage of organic soils has increased dramatically in the last decades for agricultural purposes, especialy in South-East Asia. Information on areas of drained organic soils used for agricultural or grazing purposes is therefore essential in GHG emissions assessments. A very important added value is that this study implements the Tier 1 IPCC methodology, which can be used for producing GHG inventories in the context of the global climate treaties, making it a very useful tool also for inventory preparation. No other dataset is at the moment providing this yearly information at the global level and in a spatially explicit way (although at the moment the data will be primarily distributed as country and regional statistics through FAOSTAT). Overall, this dataset is an important new part of the FAOSTAT emissions database. The paper is well written but needs some adjustments for increasing clarity. Here some suggestions beside what has been already suggested in other comments. - the authors say that their dataset covers the period 1990-2019, but the ESA-CCI timeseries cover the period 1992-2015 (then extended to 2018). Please clarify this point. - I would add some more details to the description of the datasets used. For example, to which year the livestock map refers to? How it was produced (just some details)? Which is the spatial resolution of the grid? - how is the proportion of areas of the various categories within each pixel assigned? i.e. how is the original LCCS legend used by the ESA CCI product translated to the IPCC land use categories? - Page 5 line 148:I suppose that the weighted averages refer to the Implied Emission Factors contained in the FAOSTAT dataset, not to the Emission Factors used in the estimation, which is done at pixel level and not at country level. - there are some minor language issues (e.g. page 4 line 105 "must be indeed be"), etc. so a thorough language revision would be useful.